# MindOmni: Unleashing Reasoning Generation in Vision Language Models with RGPO

**Yicheng Xiao**[1,2], **Lin Song**[2✉*], **Yukang Chen**[3], **Yingmin Luo**[2], **Yuxin Chen**[2],
**Yukang Gan**[2], **Wei Huang**[4], **Xiu Li**[1✉], **Xiaojuan Qi**[4], **Ying Shan**[2]

[1]Tsinghua Shenzhen International Graduate School, Tsinghua University
[2]ARC Lab, Tencent PCG
[3]The Chinese University of Hong Kong   [4]The University of Hong Kong
xiaoyc23@mails.tsinghua.edu.cn   ronnysong@tencent.com

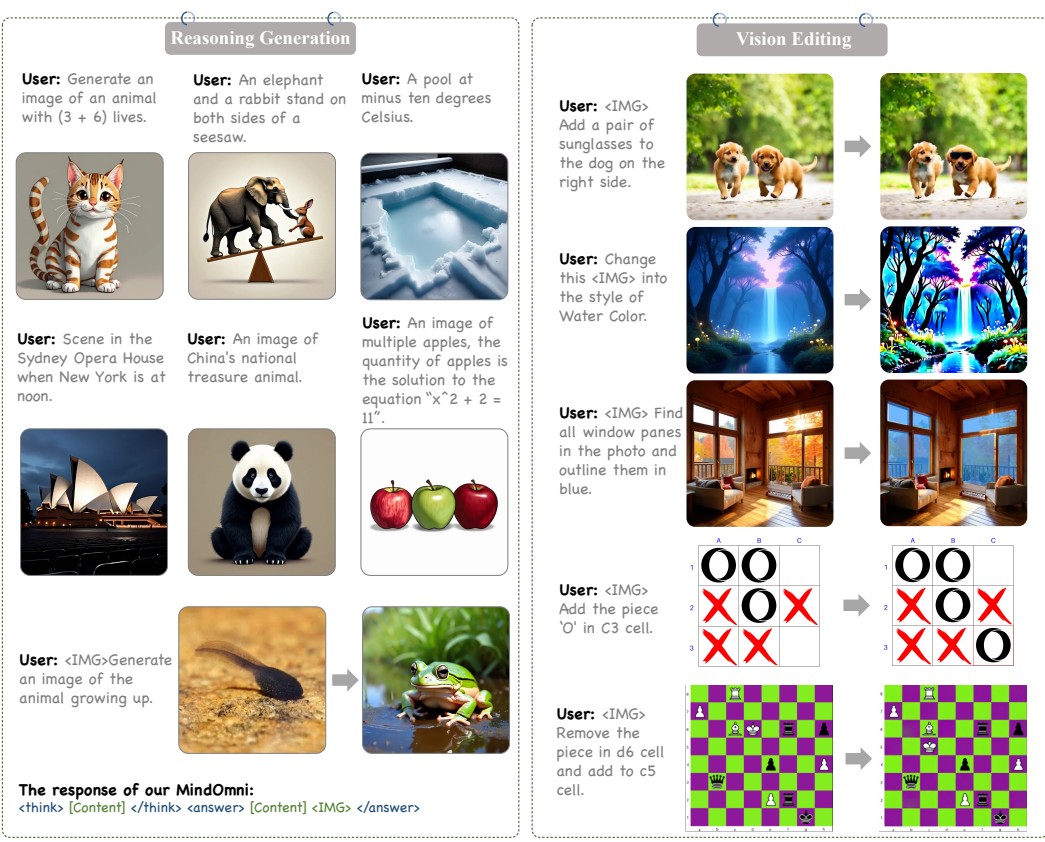

Figure 1: **Core capabilities of our proposed unified model, MindOmni.** i) Multimodal Reasoning Generation: MindOmni incorporates our proposed RGPO reinforcement learning algorithm to improve Chain-of-Thought generation, resulting in more interpretable and responsible reasoning outputs, thereby advancing reasoning generation. ii) Beyond style transfer, MindOmni preserves low-level details, such as texture, posture, and spatial layout, from the reference image, ensuring high-fidelity visual editing. Detailed reasoning outputs can be found in Appendix A.3, while a qualitative comparison with prior methods, including GPT-4o and Gemini-2.5, is presented in Fig. 6.

---

*Project Lead. ✉ Corresponding author.

39th Conference on Neural Information Processing Systems (NeurIPS 2025).

## Abstract

Recent text-to-image systems face limitations in handling multimodal inputs and complex reasoning tasks. We introduce MindOmni, a unified multimodal large language model that addresses these challenges by incorporating reasoning generation through reinforcement learning. MindOmni leverages a three-phase training strategy: i) design of a unified vision language model with a decoder-only diffusion module, ii) supervised fine-tuning with Chain-of-Thought (CoT) instruction data, and iii) our proposed Reasoning Generation Policy Optimization (RGPO) algorithm, utilizing multimodal feedback to effectively guide policy updates. Experimental results demonstrate that MindOmni outperforms existing models, achieving impressive performance on both understanding and generation benchmarks, meanwhile showcasing advanced fine-grained reasoning generation capabilities, especially with mathematical reasoning instruction. All codes will be made public at https://github.com/TencentARC/MindOmni.

## 1 Introduction

Advanced text-to-image systems such as SD-XL [32], FLUX [21], and Sana [57] generate images by leveraging textual descriptions as conditioning signals. However, they encounter two key challenges: i) The use of a fixed text encoder restricts the model to textual instructions, thereby limiting its capacity to incorporate multimodal inputs like images or audio. ii) These methods are incapable of producing explicit inferences in response to instructions that involve world knowledge, mathematical logic, or spatiotemporal perception (see Fig. 1), which is defined as reasoning generation capacity. Previous studies build unified MLLMs capable of both understanding and generation to address the first challenge. These approaches employ learnable queries [11, 8, 30] or direct projector [45] to implicitly extract semantics from VLMs, or integrate text token autoregressive and image generative processes within a single transformer [58, 64, 5] to enable early fusion. Although these approaches support multimodal inputs, they still struggle to address scenarios that require reasoning.

Building on early explorations of Chain-of-Thought (CoT) reasoning in VLMs [59, 33] and LLMs [37, 50], recent works [61, 9] propose generating textual layout plans by guiding large models with customized CoT prompts for image synthesis. While there has been progress on the second challenge, these approaches are constrained to spatial relations and hindered by intricate separation processes and inflexible CoT templates, limiting their applicability to more complex reasoning scenarios. Following the success of Deepseek-R1 [14], subsequent research [65, 4] has shown that incorporating the Group Relative Policy Optimization (GRPO) [37] algorithm into textual reasoning pipelines significantly enhances VLM performance on mathematical and coding tasks by automatically producing accurate and diverse CoT trajectories. This observation prompts a fresh perspective: *Can reinforcement learning be leveraged to unleash the reasoning generation capabilities of unified VLMs?*

To address the issue, we introduce a three-phase training strategy. In the initial stage, we propose a new multimodal large language model, named MindOmni, for unified vision understanding and generation, which includes a vision language model [1], a lightweight connector, a text head, and a decoder-only diffusion module [52]. Prior studies [14, 65] have shown that applying reinforcement learning following supervised fine-tuning can significantly enhance algorithmic performance. Therefore, we construct a series of coarse-to-fine CoT instruction data and then train our model with supervised tuning in the second stage to effectively initiate the reinforcement learning algorithm later. In the third stage, we propose the Reasoning Generation Policy Optimization (RGPO) algorithm and collect a curated corpus of plain-text reasoning sequences to guide the model in generating precise

| Method | X2Image Generation | Unified Und. & Gen. | End-to-End Pipeline | Fine-grained Image Editing | Explicit CoT | RL Augmented |
|---|---|---|---|---|---|---|
| Janus-Pro [5] | ✗ | ✔ | ✔ | ✗ | ✗ | ✗ |
| MetaMorph [45] | ✗ | ✔ | ✔ | ✗ | ✗ | ✗ |
| GoT [9] | ✔ | ✗ | ✗ | ✔ | ✔ | ✗ |
| MindOmni | ✔ | ✔ | ✔ | ✔ | ✔ | ✔ |

Table 1: **Characteristics comparison with other counterparts.**

CoT reasoning content. In contrast to the original GRPO algorithm, which relies solely on text representations, RGPO generates multimodal feedback signals from both image and text features to guide policy updates. In addition to the format reward function that enforces adherence to the correct CoT structure, we introduce a consistency reward function to evaluate visual-linguistic alignment. Tailored for reasoning generation, we introduce two independent Kullback-Leibler divergence regularizers into RGPO. They are applied separately to text and visual rollouts during policy updates, to stabilize training and mitigate knowledge forgetting. Ultimately, our MindOmni obtains obvious advantages over previous counterparts as shown in Table 1.

Fig. 1 illustrates the effectiveness of MindOmni across various complex reasoning generation scenarios, notably its strong performance on mathematical reasoning instruction. Furthermore, it showcases advanced fine-grained editing capabilities, covering general edits as well as computer vision tasks like instance segmentation. Remarkably, after fine-tuning with some specific data, MindOmni can accurately play chess and tic-tac-toe according to user inputs, establishing a foundation for future research. We also conduct extensive experiments to validate the effectiveness of MindOmni on both understanding and generation benchmarks. Our method surpasses LLaVA-One-Vision [22] by 2.4% on MMBench [62] and boosts Janus-Pro [5] by 10.6% on MMMU [63]. Furthermore, we achieve a state-of-the-art performance with 71% overall score on the reasoning generation benchmark WISE [27], which outperforms previous methods with a remarkable margin, as well as 83% overall score on the basic text-to-image generation benchmark, GenEval [12].

## 2 Related Work

### 2.1 Unified Model for Understanding and Generation

Unified multimodal LLMs have demonstrated the ability to perform both understanding and generation tasks within a single framework. Early research DreamLLM [8] and Emu-series [42, 41] predict the next multimodal element by regressing visual embeddings and integrate a separate diffusion U-Net to achieve image generation. Chameleon [43] and Emu3 [48] discretize visual features and train token-based autoregressive models on mixed image-text data. Furthermore, TransFusion [64] and Show-o [58] integrate the denoising mechanism and autoregressive approaches within a single transformer model. However, many of these unified models still fall short of the performance of task-specific architectures. In an effort to bridge this gap, Janus [51] uses distinct tokenizers for understanding and generation tasks, whereas TokenFlow [34] employs dual codebooks with shared mappings to facilitate the integration of low- and high-level features. Some methods [18, 23] aim to incorporate a unified visual tokenizer to employ a more advanced VLM, along with a separate diffusion DiT, to achieve improved performance.

### 2.2 Reasoning in Large Models

Recently, there has been a surge in research both in academia and industry regarding the reasoning capabilities of large language models. OpenAI-o1 [28] makes a significant breakthrough by enhancing reasoning performance through an extended Chain-of-Thought (CoT) reasoning framework and introducing inference-time scaling. Many studies have explored reinforcement learning [37, 20], majority voting [49], and search algorithms [19, 10] as ways to scale test-time compute and improve reasoning performance. Notably, DeepSeek-R1 [14] demonstrates exceptional reasoning performance after only a few thousand steps of RL training, leveraging the GRPO [37] algorithm. Furthermore, several studies have also applied GRPO to multimodal large models, improving visual reasoning capabilities across various tasks, including mathematical reasoning [65, 4] and reference segmentation [26].

### 2.3 Image Generation with Reasoning

Reasoning-driven image generation has made significant strides in recent years, garnering considerable attention from the academic community. PARM [15] utilizes an iterative reasoning process to enhance image quality by combining clarity and potential evaluation. RPG-DiffusionMaster [61] employs a multimodal learning model (MLLM) for text-to-image diffusion, ensuring coherence in the generated output by breaking prompts into multiple sub-regions. GoT [9] introduces a structured language reasoning framework to build a generative thinking chain, first analyzing semantic relationships and spatial arrangements, and then generating and editing images. However, GoT's

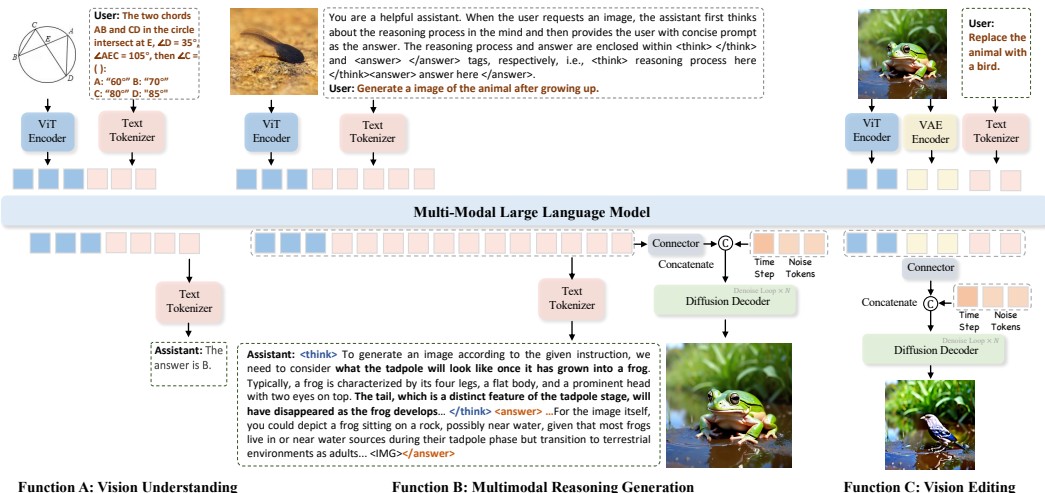

**Function A: Vision Understanding**  **Function B: Multimodal Reasoning Generation**  **Function C: Vision Editing**

Figure 2: **Overview of our inference framework.** MindOmni accomplishes vision understanding, multimodal reasoning generation, and vision editing tasks in a unified large model.

independent structure limits its flexibility and adaptability. Meanwhile, SimpleAR [47] demonstrates the effectiveness of reinforcement learning algorithms in discrete autoregressive generative models. However, previous approaches remain focused on strengthening the consistency between images and instructions using MLLM, often overlooking the use of MLLM's powerful logical reasoning capabilities for genuine reasoning generation. In this work, we aim to address this gap by introducing the RGPO reinforcement learning algorithm into a unified multimodal large model (MLLM) for both generation and understanding, enabling deeper reasoning-driven generation.

## 3 Method

In this section, we first present the pipeline formulation and describe our proposed model, MindOmni, for unified vision understanding and generation. Subsequently, we introduce our proposed Reasoning Generation Policy Optimization (RGPO) algorithm in the RL process to achieve impressive reasoning-guided image generation, with the detailed training procedure provided at the end.

### 3.1 Pipeline Formulation

A unified multimodal large model is capable of processing both visual and textual inputs simultaneously. Separate encoders map each type of input into a common text feature space. Meanwhile, a diffusion decoder conditions on the MLLM's output feature sequence to denoise the latent noise into a real image, enabling multimodal generation and vision editing. However, prior methods have primarily used multimodal models as semantic feature extractors, overlooking the inherent reasoning capabilities of LLMs. As a result, they underperform on reasoning generation and editing tasks. Recent research on the reasoning ability of large models [65, 4, 14] has inspired us to introduce reinforcement learning algorithms into our framework to unleash the reasoning generation power.

To build a unified MLLM for reasoning generation, we introduce a three-stage training pipeline and propose the RGPO reinforcement learning algorithm, which enables the model to explicitly express reasoning via a chain-of-thought process.

### 3.2 Our Model

Recent vision language models [1, 6, 22] show substantial world knowledge and cross-modal reasoning skills but lack effective generation capabilities. In contrast, existing generative models [32, 21] produce high-quality renderings but struggle to interpret complex semantic controls, such as logical reasoning or multimodal inputs. Furthermore, significant structural differences between VLMs and DiTs [31] also hinder smooth mode integration. To address it, we develop our MindOmni based on Qwen2.5-VL [1] and OmniGen [52]. The former is an advanced vision language model, and the latter

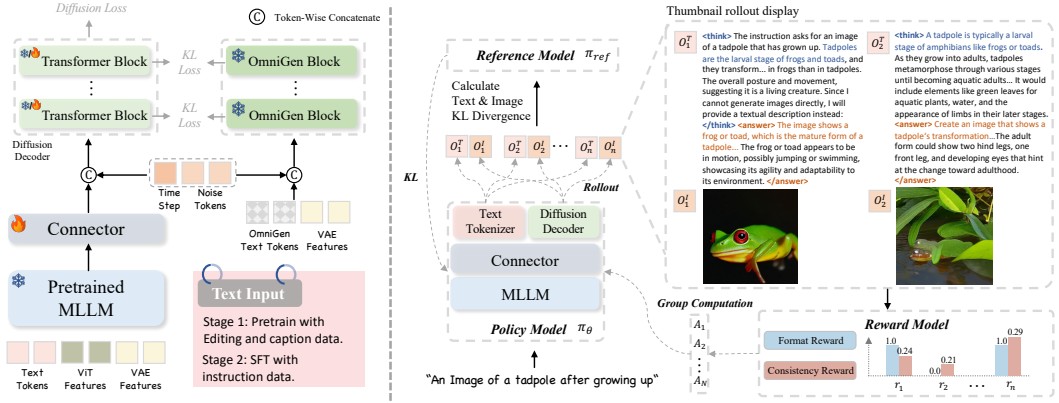

**(a) Stage 1 & Stage 2: Supervised Tuning**          **(b) Stage 3: Reinforcement Learning with RGPO**

Figure 3: **Overview of Training Pipeline.** We propose a three-stage training framework comprising pretraining, instruction-based supervised fine-tuning, and reinforcement learning with RGPO.

is a diffusion decoder that offers a promising X2Image generation performance with a streamlined, LLM-style network design. We utilize a connector comprising two standard LLM decoder layers to bridge two models. Additionally, by leveraging our RGPO reinforcement learning algorithm, MindOmni can generate more precise chains of thoughts through an autoregressive paradigm to guide reasoning generation as shown in Fig. 2.

**Vision Understanding.** Given an image $I$ and a corresponding user linguistic request for vision understanding, MindOmni first extracts continuous vision features by a pretrained ViT of Qwen2.5-VL and encodes the text into $n$ discrete tokens $T = (t_1, t_2, ..., t_n)$ with an LLM tokenizer as shown in Function A of Fig. 2. Subsequently, the model samples the output text embedding $t_{n+1}$ and detokenizes the final output according to the conditional probability:

$$P(t_1, t_2, ..., t_{n+1}|I) = \prod_{i=1}^{n+1} P(t_i|t_1, t_2, \ldots, t_{i-1}, I).$$ (1)

**Reasoning Generation and Editing.** As depicted in function B of Fig. 2, when the user provides a tadpole image and asks for a prediction of its appearance as it matures, we first capture the current visual semantics and steer the model's reasoning by adding a system prompt. The autoregressive generation process is also based on the conditional probability in Eq. (1). When the generation process encounters a special token, we use the connector to align the last hidden states of the previous tokens with the dimension of the diffusion decoder. Then we concatenate a time token (representing the time step of denoising process) $F^{\text{time}}$ along with $m$ noise latent tokens $\{F_i^{\text{noise}}|F_i^{\text{noise}} \sim \mathcal{N}(0,1)\}_{i=1}^m$ to the prior tokens $\widetilde{F^{\text{cond}}}$ which consists of vision and text features. For the vision editing task, we recognize that relying solely on ViT to extract image semantic features may cause a loss of low-level information, such as texture and layout, which can result in the inability to preserve the fine details of the reference image during editing. To address this, we incorporate the VAE features to retain these crucial details as shown in Function C of Fig. 2. At the end, the entire multimodal sequence is injected into the diffusion decoder for $N$ loops of denoising.

### 3.3 Stage 1 & Stage 2: Supervised Tuning

To implement our model, we develop a three-stage training pipeline as shown in Fig. 3. In the first two stages, we enable the model to initially achieve basic text-to-image generation and editing capabilities. In the first stage, we only train the connector using image-text pairs [39, 36] and X2I data pairs [52], ensuring that the diffusion decoder can seamlessly process the semantic representation from the MLLM. We deploy diffusion loss and KL distillation loss as objective functions to optimize our model. Following [21, 52], we use rectified flow to optimize the model. Specifically, given a ground truth image $x_0$ and a random time step $t$, we sample the noise data $\epsilon \sim \mathcal{N}(0,1)$ and get $x_t$ which is

| Type | Model | Size | GenEval | | | | | | | DPG-Bench | | |
|---|---|---|---|---|---|---|---|---|---|---|---|---|
| | | | Single Obj | Two Obj | Counting | Colors | Position | Color Attri | Overall | Global | Relation | Overall |
| *Gen. Only* | LLaMAGen [40] | 8.0B | 0.98 | 0.71 | 0.34 | 0.81 | 0.17 | 0.21 | 0.32 | - | - | 65.2 |
| | Emu3-Gen [48] | 8.0B | - | 0.81$^\dagger$ | - | - | 0.39$^\dagger$ | 0.45$^\dagger$ | 0.66$^\dagger$ | - | - | 81.6 |
| | SDv1-5 [35] | 0.9B | 0.97 | 0.38 | 0.35 | 0.76 | 0.04 | 0.06 | 0.43 | 74.6 | 73.5 | 63.2 |
| | PixArt [3] | 0.6B | 0.98 | 0.50 | 0.44 | 0.80 | 0.08 | 0.07 | 0.48 | 75.0 | 82.6 | 71.1 |
| | SDXL [32] | 2.6B | 0.98 | 0.74 | 0.39 | 0.85 | 0.15 | 0.23 | 0.55 | 83.3 | 86.6 | 74.7 |
| | SimpleAR [47] | 8.0B | - | 0.90 | - | - | 0.28 | 0.45 | 0.63 | 88.0 | 88.3 | 82.0 |
| | Flux [21] | 12B | - | - | - | - | - | - | 0.67 | - | - | 84.0 |
| | Sana-1.5 [57] | 1.6B | - | - | - | - | - | - | 0.82 | - | - | 84.5 |
| | OmniGen [52] (Baseline) | 3.8B | 0.99 | 0.86 | 0.64 | 0.85 | 0.31 | 0.55 | 0.70 | 87.5 | 88.3 | 81.2 |
| *Und. and Gen.* | Chameleon [43] | 7.0B | - | - | - | - | - | - | 0.39 | - | - | - |
| | Show-o [58] | 1.5B | 0.95 | 0.52 | 0.49 | 0.82 | 0.11 | 0.28 | 0.53 | 79.3 | 84.5 | 67.3 |
| | Janus [51] | 1.5B | 0.97 | 0.68 | 0.30 | 0.84 | 0.46 | 0.42 | 0.61 | 82.3 | 85.5 | 79.7 |
| | Janus-Pro [5] | 7.0B | 0.99 | 0.89 | 0.59 | 0.90 | 0.79 | 0.66 | 0.80 | - | - | 84.2 |
| | TokenFlow-XL[34] | 10B | 0.97 | 0.66 | 0.40 | 0.84 | 0.17 | 0.26 | 0.55 | 78.7 | 85.2 | 73.4 |
| | TokenFlow-XL[34] | 14B | 0.93$^\dagger$ | 0.72$^\dagger$ | 0.45$^\dagger$ | 0.82$^\dagger$ | 0.45$^\dagger$ | 0.42$^\dagger$ | 0.63$^\dagger$ | 78.7 | 85.2 | 73.4 |
| | GOT[9] | 7.9B | 0.99 | 0.69 | 0.67 | 0.85 | 0.34 | 0.27 | 0.64 | - | - | - |
| | MetaQuery-XL[30] | 7.0B | - | - | - | - | - | - | 0.80$^\dagger$ | - | - | 82.1 |
| | BLIP-3o$^\ddagger$[2] | 7.0B | - | - | - | - | - | - | 0.83 | - | - | 80.7 |
| | BAGEL[7] | 7.0B | - | - | - | - | - | - | 0.82 | - | - | - |
| | **MindOmni (ours)** | 7.0B | 0.99 | 0.97 | 0.77 | 0.89 | 0.59 | 0.64 | 0.81 | 89.7 | 88.7 | 83.0 |
| | **MindOmni* (ours)** | 7.0B | 0.99 | 0.94 | 0.71 | 0.90 | 0.71 | 0.71 | 0.83 | 89.1 | 89.2 | 82.5 |

Table 2: **Performance comparison on GenEval and DPG-Bench.** "Und." and "Gen." denote "understanding" and "generation", respectively. $^\dagger$ indicates using the rewritten prompts, which may improve the accuracy significantly. "Size" of unified models refers to the size of the LLM backbone. MindOmni$^*$ denotes the variant trained with higher-quality data [29, 2]. $^\ddagger$ is the released version.

defined as: $x_t = (1-t)\epsilon + tx_0$. Consequently, we denote diffusion loss as:

$$\mathcal{L}_{\text{diffuison}} = \mathbb{E}\left[\left\|\pi_\theta(x_t, t, \widetilde{F^{\text{cond}}}) - (x_0 - \epsilon)\right\|^2\right], \tag{2}$$

where $\theta$ indicates the parameters of our model and $\pi$ is our MindOmni. To mitigate the adverse effects of potential distribution shift, we employ KL divergence to enforce consistency with the teacher model $\mu$ in modeling the vector field during the denoising process. Distillation loss is formulated as:

$$\mathcal{L}_{\text{distillation}} = -\mathbb{E}\left[\log(\phi(\pi_\theta^i(x_t, t, \widetilde{F^{\text{cond}}}))\right] - \mathcal{H}(\phi(\mu^i(x_t, t, \widetilde{F^{\text{cond}}})), \tag{3}$$

where $\pi^i$ and $\mu^i$ indicate the noise latent feature of $i$-th layer in our model and teacher model, respectively. $\mathcal{H}$ represents the entropy and $\phi$ indicates the feature mapping. In the second stage, we collect captions of varying granularity to construct CoT instruction data, using coarse-grained descriptions as answer content and fine-grained descriptions as reasoning content. We also utilize leading text-to-image models [21, 57] to retrieve corresponding high-quality images. The connector and diffusion decoder are trained with diffusion loss only.

### 3.4 Stage 3: Reasoning Generation Policy Optimization

At this stage, we propose the RGPO reinforcement learning algorithm to enhance the model's reasoning generation capabilities by explicitly generating a logical chain of thought inspired by DeepSeek-R1 [14]. To achieve this, we first construct a plain-text training dataset comprising user instructions, ground-truth prompts, and corresponding explanations. Additionally, we design a reasoning generation-oriented system prompt following DeepSeek-R1 to guide our model in generating Chain-of-Thought content.

**Reward Functions.** During rollout process, policy model $\pi_\theta$ first samples $G$ groups of results $\{o_i\}_{i=1}^G$ for each request $q$, each comprising a reasoning chain $o_i^T$ and a corresponding image $o_i^I$. To enhance the quality of the generated reasoning process, we introduce two types of reward functions to guide the policy model toward producing coherent and effective outputs: i) Format reward evaluates

| Type | Model | Size | Cultural | Spatio-Temporal | | Natural Science | | | Overall |
|------|-------|------|----------|------|-------|---------|---------|-----------|---------|
| | | | | Time | Space | Biology | Physics | Chemistry | |
| *Gen. Only* | Emu3-Gen [48] | 8.0B | 0.34 | 0.45 | 0.48 | 0.41 | 0.45 | 0.27 | 0.39 |
| | SDv1-5 [35] | 0.9B | 0.34 | 0.35 | 0.32 | 0.28 | 0.29 | 0.21 | 0.32 |
| | PixArt [3] | 0.6B | 0.45 | 0.50 | 0.48 | 0.49 | 0.56 | 0.34 | 0.47 |
| | SDXL [32] | 2.6B | 0.43 | 0.48 | 0.47 | 0.44 | 0.45 | 0.27 | 0.43 |
| | FLUX [21] | 2.7B | 0.48 | 0.58 | 0.62 | 0.42 | 0.51 | 0.35 | 0.50 |
| | OmniGen [52] (Baseline) | 3.8B | 0.40 | 0.38 | 0.51 | 0.27 | 0.51 | 0.59 | 0.44 |
| *Und. and Gen.* | Chameleon [43] | 7.0B | - | - | - | - | - | - | 0.39 |
| | Show-o [58] | 1.5B | 0.28 | 0.40 | 0.48 | 0.30 | 0.46 | 0.30 | 0.35 |
| | Janus [51] | 1.5B | 0.16 | 0.26 | 0.35 | 0.28 | 0.30 | 0.14 | 0.23 |
| | Janus-Pro [5] | 7.0B | 0.30 | 0.37 | 0.49 | 0.36 | 0.42 | 0.26 | 0.35 |
| | MetaQuery-XL[30] | 7.0B | 0.56 | 0.55 | 0.62 | 0.49 | 0.63 | 0.41 | 0.55 |
| | BLIP-3o[2] | 7.0B | - | - | - | - | - | - | 0.52 |
| | BAGEL[7] | 7.0B | 0.76 | 0.69 | 0.75 | 0.65 | 0.75 | 0.58 | 0.70 |
| | **MindOmni (w/o thinking)** | 7.0B | 0.40 | 0.38 | 0.62 | 0.36 | 0.52 | 0.32 | 0.43 |
| | **MindOmni (ours)** | 7.0B | 0.60 | 0.62 | 0.64 | 0.56 | 0.68 | 0.48 | 0.60 |
| | **MindOmni* (ours)** | 7.0B | 0.75 | 0.70 | 0.76 | 0.76 | 0.72 | 0.52 | 0.71 |

Table 3: **Comparison with state-of-the-arts on WISE [27] benchmark**. "Und." and "Gen." denote "understanding" and "generation", respectively. MindOmni* denotes the variant trained with higher-quality data [29, 2] and evaluated with thinking mode.

whether the chain of thought adheres to the expected structure, returning 1 if the content is enclosed within <think> and <answer> tags, and 0 otherwise. ii) Consistency reward measures the semantic alignment between the generated image and the reference ground-truth prompt using cosine similarity derived from CLIP image and text encoders. Then the advantage $A_i$ of of the $i$-th output is computed by all reward values $\{r_i\}_{i=1}^G$, which is formulated as:

$$A_i = \frac{r_i - \bar{r}}{\text{Var}(r)}, \quad \text{while} \quad \bar{r} = \frac{1}{G}\sum_{j=1}^G r_j, \quad \text{Var}(r) = \frac{1}{G}\sum_{j=1}^G (r_j - \bar{r})^2. \tag{4}$$

**KL Regularizer.** During reinforcement learning, we incorporate two KL divergence-based distillation strategies, $\mathcal{D}_{\text{KL}}^T$ for text generation and $\mathcal{D}_{\text{KL}}^I$ for image generation to penalize large deviations between the reference model $\pi_{ref}$ and previous policy, thereby promoting smoother policy transitions and mitigating the risk of forgetting previously learned knowledge. We calculate two distillation functions of $o_i$ as:

$$\mathcal{D}_{\text{KL}}^T\left(\pi_\theta || \pi_{ref}\right) = \frac{\pi_{ref}\left(o_i^T|q\right)}{\pi_\theta\left(o_i^T|q\right)} - \log\frac{\pi_{ref}\left(o_i^T|q\right)}{\pi_\theta\left(o_i^T|q\right)} - 1, \tag{5}$$

$$\mathcal{D}_{\text{KL}}^I\left(\pi_\theta || \pi_{ref}\right) = \mathbb{E}\left[\phi(\pi_\theta(o_i^I|q))log\frac{\phi(\pi_\theta(o_i^I|q))}{\phi(\pi_{ref}(o_i^I|q))}\right]. \tag{6}$$

Finally, we optimize our policy model by minimizing the objective function $\mathcal{L}_{\text{GRPO}}$ as follows:

$$\mathcal{L}_{\text{GRPO}} = -\mathbb{E}\left[q \sim P(Q), \{o_i\}_{i=1}^G \sim \pi_{\theta_{\text{old}}}(O|q)\right]$$
$$\frac{1}{G}\sum_{i=1}^G\left(\min\left(\frac{\pi_\theta(o_i|q)}{\pi_{\theta_{\text{old}}}(o_i|q)}A_i, \text{clip}\left(\frac{\pi_\theta(o_i|q)}{\pi_{\theta_{\text{old}}}(o_i|q)}, 1-\varepsilon, 1+\varepsilon\right)A_i\right) - \beta^T\mathcal{D}_{\text{KL}}^T - \beta^I\mathcal{D}_{\text{KL}}^I\right), \tag{7}$$

where $\epsilon$, $\beta^T$ and $\beta^I$ are hyper-parameters.

## 4 Experiments

We conduct extensive experiments to evaluate the effectiveness of our MindOmni and compare it to the widely adopted large language models across diverse multimodal understanding and generation benchmarks under a fair evaluation setting.

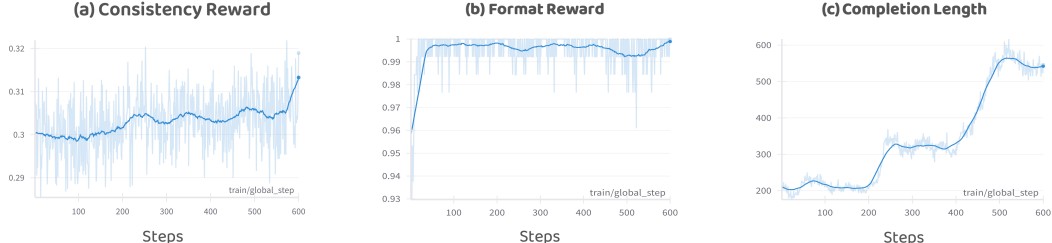

Figure 4: **Curves of different Metric in RGPO.** "Completion Length" indicates the output length of the policy model during rollout process.

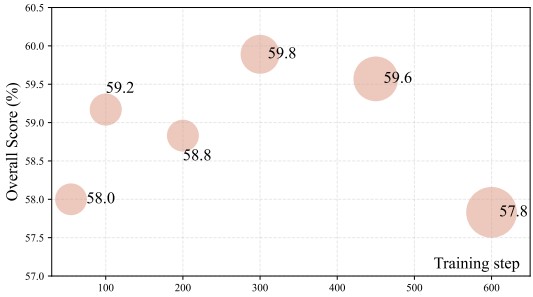

Figure 5: **Scatter plot showing the performance on WISE Benchmark.** The Y and X axes represent the overall score and training step during RL, respectively. The size of each circle reflects the completion length of our model in a certain step.

| Model | Size | MMMU | MMB | RWQA |
|---|---|---|---|---|
| *Und. only* | | | | |
| LLaVA-OV [22] | 7.0B | 48.8 | 80.8 | 66.8 |
| Emu3-Chat [48] | 8.0B | 31.6 | 58.5 | 57.4 |
| Qwen2.5-VL [1] | 7.0B | 51.1 | 83.4 | 68.9 |
| *Und. and Gen.* | | | | |
| Janus-Pro [5] | 7.0B | 41.0 | 79.2 | - |
| TokenFlow-XL [34] | 14B | 38.7 | 68.9 | 58.3 |
| MetaMorph [45] | 8.0B | 41.8 | 75.2 | 68.8 |
| **MindOmni (ours)** | 7.0B | 51.6 | 83.2 | 68.1 |

Table 4: **Performance Comparison on Vision Understanding Benchmarks.** "Und." and "Gen." denote "understanding" and "generation," respectively.

### 4.1 Training Details

We utilize the open-sourced image-caption pairs [36, 39] and in-house data as training data. Group Number $G$, KL coefficient $\beta^I$ and $\beta^T$ in RGPO are set as 4, 0.06 and 0.01 by default. Due to the limited space, detailed contents are present in Appendix A.1.

### 4.2 Main Results

**Image Understanding and Generation.** Our performance on image understanding is shown in Table 4. We first evaluate our MindOmni on understanding benchmarks (MMMU [63], MMBench [62] and RealworldQA). Due to the inclusion of the text KL regularization term, the model's performance after reinforcement learning fine-tuning remains nearly identical to the original VLM backbone, with only a 0.1% average difference. MindOmni boots previous unified counterparts, Janus-Pro by 10.6% on MMMU and MetaMorph by 9.8% on MMBench. Furthermore, we validate our common text-to-image generation ability on GenEval [12] and DPG-Bench [16]. MindOmni achieves state-of-the-art performance on GenEval in unified MLLM with 83% overall score as shown in Table 2. We build MindOmni based on OmniGen [52], which outperforms MetaQuery [30] with Sana [57] as the generator on DPG-Bench by 0.4% as shown in Table 3.

**Reasoning Generation.** To evaluate the reasoning-aware generation ability of our proposed MindOmni, we benchmark it against a broad set of existing generative-only and unified models using a comprehensive evaluation suite [27] that spans cultural knowledge, spatio-temporal reasoning (time and space), and natural science domains (biology, physics, and chemistry). As shown in Table 3, MindOmni outperforms all methods across nearly all reasoning subcategories, achieving an overall score of 0.71 with the thinking mode. Specifically, MindOmni surpasses prior unified models such as MetaQuery-XL [30] (0.55 overall) by notable margins in both time reasoning and physics reasoning. Compared to leading generative-only models such as FLUX (0.50 overall) and PixArt (0.47 overall), MindOmni demonstrates consistent advantages across every category, reflecting the benefit of joint understanding-generation training and CoT-guided reinforcement learning. Additionally, Fig. 4

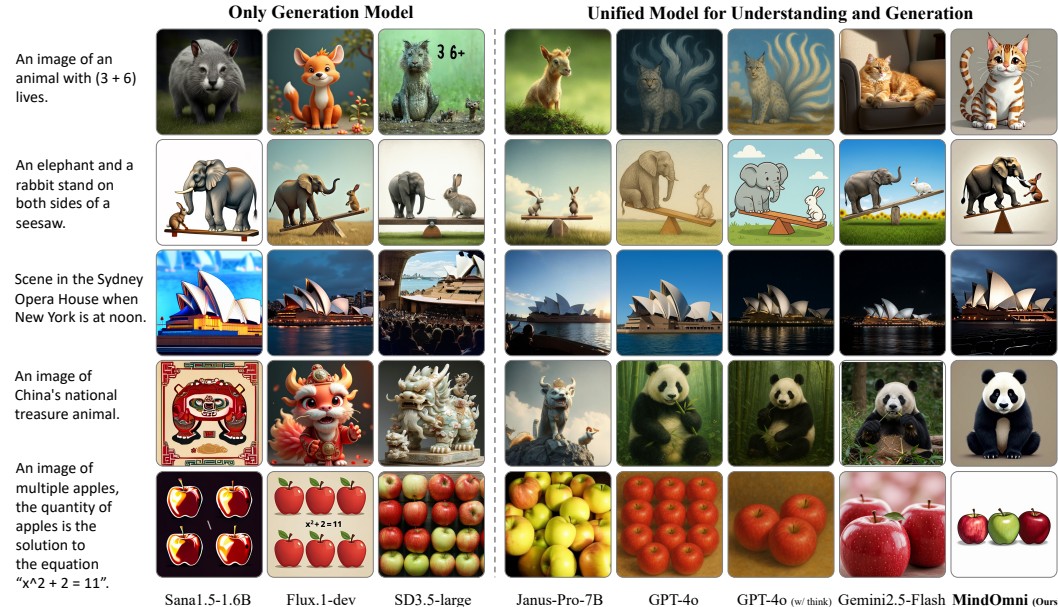

Figure 6: **Qualitative comparison among leading models on reasoning-aware image generation.**

| Stage 1 | Stage 2 | Stage 3 | GenEval | WISE |
|:---:|:---:|:---:|:---:|:---:|
| ✓ | | | 0.73 | 0.42 |
| ✓ | ✓ | | 0.81 | 0.54 |
| ✓ | | ✓ | 0.72 | 0.49 |
| ✓ | ✓ | ✓ | 0.81 | 0.60 |

Table 5: **Effectiveness of the training stages.**

| Type | Layer Number | Param. | #FLOPs | GenEval |
|:---:|:---:|:---:|:---:|:---:|
| MLP | 1 | 0.1% | 2G | 0.63 |
| Decoder | 1 | 1.5% | 31G | 0.68 |
| | 2 | 3.0% | 60G | 0.73 |

Table 6: **Impact of different connectors.**

illustrates the progression of various metrics throughout the reinforcement learning process, with steady increases in both the consistency reward and completion length. However, as shown in Fig. 5, merely increasing the length of output thinking content does not lead to better end results.

## 4.3 Qualitative Results

We visualize and compare advanced text-to-image methods [21, 38, 57] and unified systems on the reasoning generation task, including GPT-4o [29] and Gemini-2.5 [13] as depicted in Fig. 6 and Fig. 7. In the absence of a reasoning mode, GPT-4o is unable to perform reasoning generation for scenarios involving mathematical logic and physical perception. More basic text-to-image results are shown in Fig. 8 in Appendix.

## 4.4 Ablation Study

**Effectiveness of Training Stage.** We conduct an ablation study to assess the contribution of each training stage. As shown in Table 5, stage 1 (pre-training) initially expands the generation capability of MindOmni. Adding Stage 2 (thinking template SFT) brings a significant boost, especially on WISE (+0.12). In contrast, skipping Stage 2 and applying RGPO alone leads to a drop in GenEval (0.72) and a smaller WISE boost (0.49), suggesting the necessity of CoT instruction SFT.

**Impact of Different Connectors.** We briefly compare 2 types of projectors and find that using the decoder projector can better convey the semantic from VLM.

**Ablation on RGPO.** We evaluate our algorithm's performance under various settings for the RGPO stage, including the KL coefficient, group numbers, and reward strategies as shown in Table 7, Table 8 and Table 9, respectively. Consistency reward plays a key role in steering reasoning generation, and

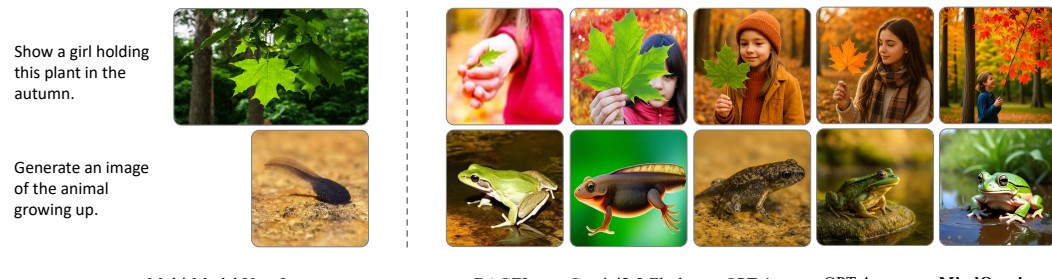

Figure 7: **Qualitative comparison among leading models on reasoning-aware image generation with multimodal user input.**

| $\beta^T$ | $\beta^I$ | GenEval | WISE |
|---|---|---|---|
| 0 | 0.01 | 0.79 | 58.8 |
| 0.01 | 0.06 | 0.81 | 59.8 |
| 0.004 | 0.01 | 0.80 | 58.9 |
| 0.01 | 0 | 0.78 | 58.2 |

Table 7: **Effectiveness of different KL coefficient.**

| Group Number | GenEval | WISE |
|---|---|---|
| 2 | 0.79 | 59.2 |
| 4 | 0.81 | 59.8 |
| 6 | 0.79 | 59.3 |

Table 8: **Effectiveness of different group numbers.**

| Format Reward | Consistency Reward | WISE |
|---|---|---|
| ✓ | | 56.1 |
| | ✓ | 59.4 |
| ✓ | ✓ | 59.8 |

Table 9: **Effectiveness of different reward strategies.**

the inclusion of KL divergence for the visual modality helps the model maintain its fundamental text-to-image generation abilities.

## 5    Conclusion

In this paper, we introduce MindOmni, a unified MLLM that enhances reasoning generation through reinforcement learning. Our three-phase training strategy significantly achieves impressive performance on both vision understanding and text-to-image generation tasks, outperforming existing counterparts. Furthermore, MindOmni demonstrates strong reasoning-aware generation capabilities, including mathematical problem-solving and time-space perception, paving the way for future advancements in multimodal AI systems.

## Acknowledgments and Disclosure of Funding

This work was partly supported by Shenzhen Key Laboratory of next generation interactive media innovative technology (No:ZDSYS20210623092001004) and National Natural Science Foundation of China (No.62293544, 62425117).

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

# A    Appendix

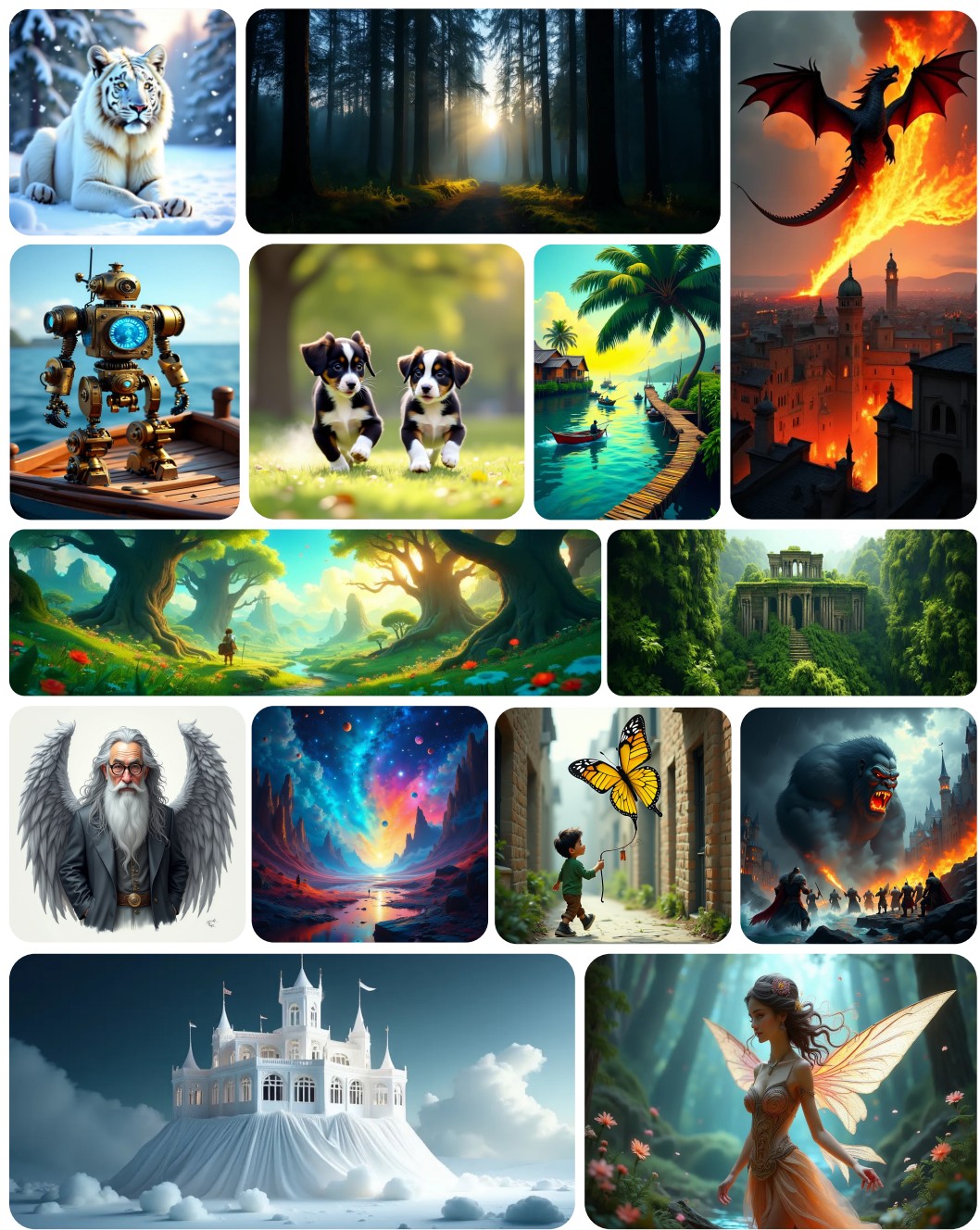

Figure 8: **More qualitative results across different resolutions.**

## A.1    Implementary Details

In the first stage of training, we utilize the Laion COCO [36] and JourneyDB [39] image-caption pairs as foundational text2image data. Additionally, we incorporate the X2I dataset [52], which includes tasks for computer vision [54], in-context learning, multi-modal instructions, and subject-driven. To further enrich our dataset, we leverage Sana [57] and Flux [21] to generate high-quality images based on the existing prompt. During training, we use a constant learning rate scheduler with an initial rate of $1 \times 10^{-4}$ and a weight decay of 0.05. The model is trained with a batch size of 1024 using

images at a resolution of $256 \times 256$. In the second stage, we employ Qwen2.5-VL [1] to generate CoT instruction data. At this stage, we progressively increase the training resolution to $512 \times 512$, while reducing the learning rate to $5 \times 10^{-5}$. In the final stage, we utilize Qwen3 [44] to generate a dataset of logical reasoning texts, which serve as training data for reinforcement learning. We adopt a cosine scheduler following previous works [52, 53, 55] during the reinforcement learning phase.

We evaluate our method during multimodal understanding and various generation benchmarks. Specifically, our model is evaluated on MMMU [63], MMBench [62] and RealworldQA for image understanding. As for basic image generation tasks, we evaluate our model on GenEval [12], which involves various metrics such as counting, colors, and position. We also evaluate our text-to-image generation capability on DPG-Bench [16] following previous methods [5, 34]. For reasoning generation evaluation on WISE [27] Benchmark, we perform our MindOmni into thinking mode and generate the corresponding image with $1024 \times 1024$.

## A.2 Image Editing Performance

We conduct evaluations on GEdit [25] to verify the performance on image editing task of our method. In detail, under the fair setting as previous models [25, 66], our model achieved SC and PQ scores of 6.55 and 6.92, respectively, with an overall accuracy of 5.95.

## A.3 Detailed Response of Our MindOmni

For the user input of different reasoning generations in Fig. 1, the detailed response of our model MindOmni is shown in Fig. 9, Fig. 10, Fig. 11 and Fig. 12.

**User:** Generate an image of an animal with (3 + 6) lives.
**MindOmni: <think>** To generate an image based on the instruction "An image of an animal with (3 + 6) lives," we first need to interpret the mathematical expression given: **\\(3 + 6 = 9\\).** This suggests that the animal should have nine lives, which is a common characteristic associated with mythical creatures like cats or dragons.

**Given this interpretation, I would suggest creating an image of a cat, which is known for having nine lives.** The image could be a digital drawing or illustration of a cat, possibly with a playful and whimsical style to reflect its enduring and resilient nature.
The instruction does not specify the type of animal further, but commonly, a cat is used when referring to the concept of "nine lives" due to the popular belief that a cat can survive falls from great heights, often attributed to its ability to land on its feet.
For the image:
- The cat could be depicted in a simple yet vibrant drawing.
- It might be shown in a dynamic pose, perhaps mid-leap or after a sudden turn.
- The cat could have a light, fluffy appearance with large eyes and a small body, capturing the classic cartoonish representation of a cat.

This approach aligns with the whimsical and imaginative nature typically associated with the idea of a cat having nine lives.**</think>**

**<answer>** To create an image according to the instructions, you could generate a digital drawing or illustration of a cat. The cat should have a playful and whimsical style, possibly depicted in a dynamic pose such as mid-leap or after a sudden turn.

The cat could be drawn with a light, fluffy appearance, large eyes, and a small body, capturing the classic cartoonish representation of a cat. This image would visually represent the concept of the cat having nine lives, as traditionally associated with feline resilience and agility. ****
**</answer>**

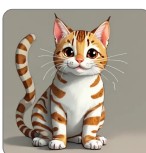

**User:** An elephant and a rabbit stand on both sides of a seesaw.
**MindOmni: <think>** To generate an illustration of an elephant and a rabbit standing on both sides of a seesaw, we need to consider a few key elements: the size difference between the elephant and the rabbit, the balance and movement implied by the seesaw, and the positioning of the animals. Elephants are much larger than rabbits, so they would be on one end of the seesaw, while the rabbit would likely be at the other, opposite end.

The seesaw is typically represented as a horizontal bar balanced on a central fulcrum. The elephants and the rabbit would each be standing on a platform or platform-like structure that extends from either side of this central bar. **The elephant's end would be lower because its mass is significantly greater than the rabbit's. The rabbit's end would likely be higher due to the balancing effect of the seesaw.**

In terms of style, the illustration could be cartoonish for a simple, child-friendly look, or more realistic if aiming for a detailed and lifelike depiction. The background can be minimal, perhaps just suggesting an open space, which keeps the focus on the main subjects.

To create such an image, one might start with a simple sketch, then add details like the elephant's trunk, the rabbit's long ears, the seesaw's balance point, and the platforms. Color choices could include earthy tones for the elephant (like brown) and lighter, more playful colors for the rabbit (like pink or yellow). **</think>**
**<answer>** Creating an image of an elephant and a rabbit on a seesaw involves drawing or digitally rendering two large elephants and a small rabbit placed at opposite ends of a seesaw. The elephant should be standing on a platform near one end of the seesaw, and the rabbit should be on the opposite end. The seesaw should be shown balancing in the middle point, with the elephant's end appearing lower due to its larger mass and the rabbit's end appearing higher.

For a simple illustration:
1. Sketch the outline of the seesaw with the fulcrum in the center.
2. Add the elephant on one side with its trunk possibly hanging down or interacting, and the rabbit on the other side.
3. Color the elephant in darker shades, the rabbit in lighter, playful colors.
4. Ensure the seesaw is balanced correctly, with the elephant's end lower and the rabbit's end. ****
**</answer>**

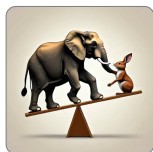

Figure 9: **Detailed response of our MindOmni.**

**User:** A pool at minus ten degrees Celsius.

**MindOmni: \<think\>** To generate an image of a pool at minus ten degrees Celsius, **one should consider the freezing point of water (0°C or 32°F) and the consequences of the water freezing below this temperature.** At minus ten degrees Celsius, water would indeed be frozen into ice, forming a solid surface. The appearance would likely include a clear, uncolored ice layer covering the pool's water, which, due to the cold, would have a glassy, possibly cloudy texture. The image would depict a scene where the entire body of water is frozen, creating a hard, icy surface on top.

To create such an image, it might involve drawing or digitally rendering a rectangular or circular pool shape filled with a smooth, flat, transparent ice layer. The background could show the outline of the pool with visible edges, suggesting a boundary for the frozen area. The ice surface can be depicted in a light blue or white color, with subtle shading to indicate the thickness of the ice and any slight irregularities that could be present in real-life scenarios. The image should not include any liquid water, as at minus ten degrees Celsius, the water has already frozen.

The visual representation will be static, as the image would not capture movement; it would focus solely on the frozen state of the water, highlighting the transformation from liquid to solid under extreme cold conditions.**\</think\>**

**\<answer\>** To create the image, start by drawing a rectangular or circular outline representing the pool. Fill the pool with a flat, transparent ice layer that covers the entire surface. The ice can be rendered in a light blue or white color, with slight variations in hue to suggest thickness and the potential for small air bubbles or impurities trapped in the ice. Ensure the ice layer is smooth but shows some variation in texture to reflect its real-world appearance.

The overall image should emphasize the immobility and stillness of the frozen water, with no indication of liquid water beneath the ice. The icy surface should be depicted as a hard, solid cover over the entire pool area, with no thawing or melting visible in the image. The environment surrounding the pool should be kept neutral to avoid drawing attention away from the main subject—the frozen pool. **\ \</answer\>**

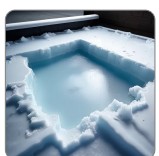

**User:** Scene in the Sydney Opera House when New York is at noon.

**MindOmni: \<think\>**To generate an image of the Sydney Opera House when New York is at noon, we need to consider the geographical and time difference between Sydney (which is in the Eastern Australia Time Zone) and New York (which is in Eastern Standard Time). **The time difference between these two locations is approximately 15 hours, with New York being 15 hours ahead of Sydney. Therefore, when it's noon in New York, it would be approximately 3 AM in Sydney.**

The image should reflect the interior of the Sydney Opera House during this late hour of the night or early morning. The distinctive curved shell-like structures characteristic of the Opera House would likely be illuminated by the artificial lighting inside the building, which would contrast with the dark surroundings outside. The seating areas, corridors, and architectural features would be highlighted.

Given that the specific time would be close to dawn, there might be a subtle hint of light or the appearance of sunrise through windows or doors leading to the interior.

In terms of capturing this scene artistically, the focus would be on the interplay of shadows and light within the interior design of the Opera House, emphasizing the futuristic and modern aesthetic of the structure. The overall atmosphere would be calm and serene, with the darkness outside juxtaposed against the bright interior lights.**\</think\>**

**\<answer\>** This could be best conveyed through a digital illustration or a photo composition. You might use software like Adobe Photoshop or digital drawing tools where you draw or composite elements of the interior of the Sydney Opera House, specifically highlighting its iconic architectural features like the white curves and the rows of seating, all rendered inside the building.

The scene would be set in a still, minimalist environment mimicking 3 AM, with ambient lighting creating soft patterns across the surfaces of the seats and the ceiling. You could add a faint glow from the windows showing the beginning of daylight outside to provide a sense of the time. The lighting within the Opera House would depict a variety of light sources: warm accent lights for dramatic effect, and the soft, scattered lights from a dim evening ambiance. **\ \</answer\>**

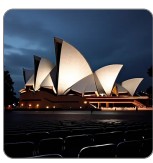

Figure 10: **Detailed response of our MindOmni.**

## A.4  Limitation&Future Work

In this work, our Chain-of-Thought (CoT) content is presented in plain text, while multimodal CoT processes will be addressed in future research. Furthermore, enhancing reasoning efficiency [24] and expanding the unified model to support additional modalities and tasks [56, 17, 60, 66, 46] represent promising prospects for future investigation.

## A.5  More Qualitatvie Results

We provide more qualitative results about text2image generation of our MindOmni as shown in Fig. 8.

**User:** An image of China's national treasure animal.

**MindOmni:** **\<think\>** To generate an image of China's national treasure animal, one should consider which animal is commonly recognized as the "national treasure" in Chinese culture. **This is often the giant panda (Ailuropoda melanoleuca). Pandas are iconic symbols of China and are known for their distinctive black-and-white fur pattern.**

The instruction does not specify the medium or style of the image, so we can create a digital illustration of a panda. Pandas typically have a round face, large dark eyes, and a body that is mostly white with black markings around their eyes, ears, legs, and shoulders.

For the style, a simple cartoon-like or realistic style could be used, depending on the intended audience and context. If the image is meant to be educational or for children, a cartoonish style might be more appropriate. For a more formal or artistic representation, a more detailed and realistic depiction would be suitable**.\</think\>**

**\<answer\>** To create an image of China's national treasure animal, such as a giant panda, you could start by drawing the characteristic features: a round face, large dark eyes, and a body that is predominantly white with black patterns around the eyes, ears, legs, and shoulders. A simple cartoon style or a more detailed, realistic rendering can both capture the essence of a panda**. \ \</answer\>**

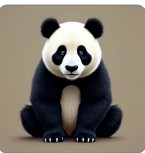

**User:** An image of multiple apples, the quantity of apples is the solution to the equation "x^2 + 2 = 11".

**MindOmni:** **\<think\>**To generate the image, we need to solve the given equation for x first: \\(x^2 + 2 = 11\\). Subtracting 2 from both sides gives us \\(x^2 = 9\\). **Taking the square root of both sides, we get \\(x = \\pm3\\). Since the number of apples must be a positive integer, we consider only the positive solution, which is 3. Therefore, the image should depict three apples.**

The next step is to create an image that visually represents this solution. This could be a simple drawing or a digital illustration showing three apples. The apples can be depicted in any style, such as a cartoonish or realistic representation, but the key point is that there should be exactly three apples in the image.**\</think\>**

**\<answer\>** To create the image, you would draw or digitally render three apples. The apples can be placed in a row, stacked, or arranged in any way that clearly shows the number three. The image should be simple enough to convey the solution to the equation \\(x^2 + 2 = 11\\) by visually representing the quantity of apples, which is 3.

You might use a white background with the apples in a natural color like red or green, depending on your preference. If you're using a digital tool, you can draw the apples directly or use clip art to place them in the image. The focus should be on clarity and simplicity to ensure the viewer can easily understand that the image represents the solution to the equation, which is three apples. \ **\</answer\>**

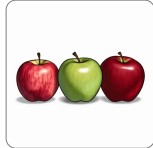

Figure 11: **Detailed response of our MindOmni.**

**User:** \<IMG1\> Generate an image of the animal growing up.

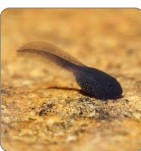

**MindOmni:** **\<think\>** To generate an image according to the given instruction, we need to consider what the tadpole will look like once it has grown into a frog. **Typically, a frog is characterized by its four legs, a flat body, and a prominent head with two eyes on top. The tail, which is a distinct feature of the tadpole stage, will have disappeared as the frog develops**. The transition from tadpole to frog involves significant physical changes that include the development of hind legs before the front legs, the growth of the lungs for breathing air, and the transformation of the tail into hind legs. The skin also thickens, becoming smoother and drier.

Given the instruction to generate an image of the animal after growing up, I will create a depiction of a fully developed frog, which is the adult form of the frog species depicted here. **\</think\>\<answer\>** To generate an image of a frog after the tadpole stage, you might imagine a frog with a robust body, smooth and dry skin, and four sturdy legs, including hind legs that are well-developed and strong enough to support the frog's weight. The frog's head would be large and round, with bulging eyes. It would likely have a streamlined body shape and a broad snout with a moist appearance.

For the image itself, you could depict a frog sitting on a rock, possibly near water, given that most frogs live in or near water sources during their tadpole phase but transition to terrestrial environments as adults. The frog's coloration can vary greatly depending on the species, but many frogs have patterns that help them blend into their environment, such as green, brown, or spotted patterns.

Since you didn't specify any particular species of frog, the image could represent a generic frog, which would typically have these characteristic features.

Please note that this description is a general representation based on common frog characteristics. If you're looking for a specific species, additional details about the frog's appearance, such as coloration and pattern, would be necessary to accurately depict the frog. **\<IMG2\>\</answer\>**

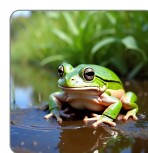

Figure 12: **Detailed response of our MindOmni.**

