# OpenReview forum: "MindOmni: Unleashing Reasoning Generation in Vision Language Models with RGPO"
_NeurIPS.cc/2025/Conference — NeurIPS 2025 poster_

### Official Review · Reviewer_Si6L · 2025-07-02

**Clarity:** 3
**Significance:** 3
**Originality:** 3
**Rating:** 4
**Confidence:** 5

**Summary:**

The paper presents MindOmni, a unified multimodal large language model that enhances reasoning generation in vision-language models via the proposed Reasoning Generation Policy Optimization (RGPO) algorithm. It employs a three-phase training strategy, combining a unified vision-language model with a decoder-only diffusion module, supervised fine-tuning with Chain-of-Thought data, and RGPO to leverage multimodal feedback for policy updates. Experiments demonstrate that MindOmni achieves impressive performance on both understanding and generation benchmarks, particularly in mathematical reasoning and spatial-temporal tasks, demonstrating its advanced fine-grained reasoning generation and image editing capabilities. The work advances the field by enabling deeper reasoning-driven generation in unified multimodal models, bridging the gap between vision understanding and complex reasoning tasks.

**Questions:**

* The claim that the model can "accurately play chess" is a particularly strong statement. However, the paper presents no experimental results, evaluation protocols, or qualitative/quantitative evidence to support this assertion. Additional experiments and analyses are needed to substantiate this claim.

*  Policy collapse, entropy collapse, and reward collapse are common issues during RLVR training. Although the authors propose a dual KL-divergence regularization technique as an interesting stabilization method, the paper lacks an in-depth analysis of its effectiveness, such as a thorough comparison with standard single-KL regularization. Moreover, the phenomenon of "overthinking" revealed in the results is not discussed in the context of existing literature.

**Ethical Concerns:**

["NO or VERY MINOR ethics concerns only"]

**Final Justification:**

The authors have clarified the issues regarding the acquisition of Chain-of-Thought abilities and the details of the RLVR reward. I recommend this paper for acceptance.

**Limitations:**

Yes

**Paper Formatting Concerns:**

There are no apparent formatting issues.

**Quality:**

3

**Strengths And Weaknesses:**

**Strengths**
* This paper is clearly written and easy to follow.
* The core contribution of this work lies in the application of a class of policy gradient reinforcement learning algorithms—specifically, the proposed RGPO—to unified multimodal generation models. By explicitly modeling and optimizing the CoT reasoning process, the authors provide a strong baseline for enhancing the controllability and complex instruction-following abilities of multimodal models. Furthermore, this approach offers potential insights for the development of o4-like models and world models.

**Weaknesses**
* The consistency reward based on CLIP similarity represents a major limitation of RG-VLM. Extensive prior work has demonstrated that CLIP performs poorly in evaluating complex spatial, physical, or logical relationships. For example, while CLIP can recognize objects within an image, it often fails to verify the correctness of their spatial arrangements. As a result, the reward signal serves as a very coarse and unreliable proxy for reasoning ability. An unhackable and trustworthy reward is arguably the most crucial factor for RLVR training effectiveness and stability.

* In the critical second stage of the pipeline, the paper uses another model (Qwen2.5-VL) to synthesize CoT data, yet provides no description of its quality control, diversity assurance, or bias filtering mechanisms. This raises concerns that the model’s reasoning ability may be inherently limited by the teacher model’s upper bound, and that it may end up merely imitating the specific "thinking style" of the teacher model rather than acquiring generalized reasoning skills. In current industry practice, there are primarily two cold-start strategies: distillation or zero-shot self-bootstrapping followed by filtering. If the CoT data is generated by a model lacking genuine reasoning capacity, it is unlikely to exhibit robust reasoning patterns.

---

> ### Author Rebuttal · Authors · 2025-07-31
>
> We begin by sincerely appreciating the reviewer for the respectful and patient feedback. We completely understand the concerns raised and will address them individually for further clarification.
>
> **W1: Question about the CLIP reward.**
>
> **Ans:**
> We agree with the reviewer that CLIP-based similarity metrics have known limitations in capturing complex spatial or logical relationships. However, our main contributions are the proposal of a unified reasoning-generation model and a training RL framework that enhances its reasoning generation ability. While we initially chose CLIP rewards to discourage benchmark hacking, we also explored a more capable reward model that better aligns with fine-grained reasoning. This alternative yielded a 0.8-point performance improvement under the same settings, as shown in Table 13 (in the response to reviewer Joq5’s W3).
>
> **W2: Question about the training stage 2.**
>
> **Ans:**
> On the one hand, the CoT data in stage 2 is composed of short, coarse image captions as input prompts, with the corresponding “reasoning” content being detailed and descriptive long-form captions, which are generated by Qwen2.5VL-72B. On the other hand, we utilize this data to train only the generation decoder, enabling it to adapt specifically to Chain-of-Thought contextual inputs. The VLM component remains frozen throughout this phase, thereby preserving its general reasoning capabilities and ensuring that the model is not constrained by the limitations of the additional model. If this stage is omitted, the model's performance on the WISE benchmark will drop from 60% to 49% as shown in Table 5 in the manuscript.
>
> **Q1: The claim that the model can ”accurately play chess” is a particularly strong statement.**
>
> **Ans:**
> We regret any potential misunderstanding. Our goal was to underscore the model’s ability to perform fine-grained editing tasks, such as manipulating chess pieces according to specific instructions. We create a chessboard test dataset to assess this functionality and also evaluate the model on public editing benchmarks (GEdit and ImgEdit) as shown in Table 15 below. This fine-grained control is essential for enabling future research into adaptive chess playing. We will include these experimental results and the corresponding benchmark descriptions in the revised version.
>
> |                | **Chess** | **GEdit** | **ImgEdit** |
> |----------------|:------------:|:----------:|:--------:|
> | baseline       |      0.45    |    0.503   |   0.296  |
> | RG-VLM (ours)  |      0.52    |    0.599   |   0.333  |
>
> **Table 15: Performance of editing task.**
>
> **Q2: Question about training stage 3.**
>
> **Ans:**
> We have provided extensive ablation studies in Tables 7–9 of the manuscript, evaluating different aspects of the RGPO reinforcement learning strategy, including KL regularization schemes, group numbers, and reward strategies. Specifically, Table 7 shows that standard single-KL term ($\beta^{I}$=0) reduces performance by 1.6 points on the WISE benchmark, underscoring the effectiveness of our dual KL approach. Furthermore, Figure 4 illustrates stable reward behavior throughout training, suggesting that KL regularization effectively mitigates reward collapse. We acknowledge the overthinking phenomenon observed during our RL process and note that it aligns with findings in prior works [1,2]. We will incorporate a discussion of this effect and relevant citations in the revision.
>
> [1] Mitigating Overthinking in Large Reasoning Models via Manifold Steering
>
> [2] Think How to Think: Mitigating Overthinking with Autonomous Difficulty Cognition in Large Reasoning Models
>
> Please let us know if you have any unsolved or other concerns.

---

> > ### Author Response · Authors · 2025-08-05
> > **Discussion Request with reviewer Si6L**
> >
> > Dear Reviewer Si6L,
> >
> > We sincerely appreciate your valuable review time and comments! As the discussion deadline approaches, we kindly wonder if our response has sufficiently addressed your concerns. Please feel free to share any further feedback, and we will gladly discuss it and provide additional clarification.
> >
> > Best regards,
> >
> > Paper 13149 Authors

---

> > > ### Comment · Reviewer_Si6L · 2025-08-05
> > > **The comments to the authors' responses**
> > >
> > > Thank you for your response and the detailed supplementary experiments. Your answers have addressed most of my concerns. However, I still have questions regarding the cold start data:
> > >
> > > I am concerned about the quality of the synthetic CoT data and the method you mentioned in your third stage, which uses DeepseekR1 to guide the model's reasoning. I believe these methods are crucial. Additionally, I would like to learn more about the amount of data used in your RL stage. Addressing these concerns would lead me to consider revising my score.

---

> > > > ### Author Response · Authors · 2025-08-05
> > > > **Response to Reviewer Si6L**
> > > >
> > > > Dear Reviewer Si6L:
> > > >
> > > > We firstly appreciate the opportunity you provided to make further clarification and are pleased that we have addressed most of your concerns. Our responses are organized as follows:
> > > >
> > > > 1. **About 'CoT' Data:** A potential misunderstanding may arise at this point. **First, it is important to clarify that the CoT data in Stage 2 is not used to train the VLM. We froze the VLM during this stage to preserve its original reasoning ability and aim to transfer that ability to the image generation process.** Moreover, during the entire training process, we did not construct any additional CoT data except in Stage 2. Specifically, as described in Line 163 of the manuscript, during the second training stage, we employ Qwen2.5VL-72B to rewrite the descriptions of image-text pairs, producing two types of captions: fine-grained (long, detailed descriptions of visual elements) and coarse-grained (short, concise summaries). The training data format is: ``<user><request><assistant><think><long caption></think><answer><short caption><generated image></answer>``. This data structure helps the diffusion decoder better adapt to the reasoning format, which is beneficial for stage 3. As shown in Table 16 below (as the same in Table 5 of the manuscript), omitting stage-2 results in an 11.5 point performance degradation.
> > > >
> > > > | **Stage 1** |**Stage 2**  | **Stage 3** | **WISE (Reasoning Image Generation Benchmark)** |
> > > > |:--------:|:-------------:|:---------:|:-------:|
> > > > |√ | - | - | 42.1 |
> > > > |√ | √ | - | 54.4 |
> > > > |√ | - | √ | 49.2 |
> > > > |√ | √ | √ | **60.7** |
> > > >
> > > > **Table 16: Ablation of different training stages.**
> > > >
> > > > 2. **Stage-3 Clarification:** Actually, we did not rely on DeepSeek-R1 for direct supervision. Instead, we introduced RGPO, a reinforcement learning algorithm to promote the VLM’s spontaneous CoT generation. We train VLM in this stage and the VLM policy is optimized with rewards based on image-text consistency and thinking format adherence as illustrated in Line 174 of the manuscript. This stage yields notable improvements with a 6.3 point increase in image reasoning generation performance, as shown in Table 16.
> > > >
> > > > 3. **Stage-3 Data Volume:** This stage involved 19k reasoning training samples over 600 iterations, consistent with the scale adopted in reinforcement learning studies[1][2][3].
> > > >
> > > > Thank you again for your reply. If any concerns remain unaddressed, we would be glad to make further clarification!
> > > >
> > > > Best regards,
> > > >
> > > > Paper 13149 Authors
> > > >
> > > > [1] Guo, Daya, et al. "Deepseek-r1: Incentivizing reasoning capability in llms via reinforcement learning." arXiv preprint arXiv:2501.12948 (2025).
> > > >
> > > > [2] Meng, Fanqing, et al. "Mm-eureka: Exploring visual aha moment with rule-based large-scale reinforcement learning." CoRR (2025).
> > > >
> > > > [3] Shen, Haozhan, et al. "Vlm-r1: A stable and generalizable r1-style large vision-language model." arXiv preprint arXiv:2504.07615 (2025).

---

> > > > > ### Comment · Reviewer_Si6L · 2025-08-07
> > > > >
> > > > > The authors have addressed my concerns, and I believe this paper represents a very interesting exploration of unified MLLMs in the domain of reasoning. Therefore, I am raising my score and look forward to the authors' future impactful work.

---

> > > > > > ### Author Response · Authors · 2025-08-07
> > > > > > **Response to Reviewer Si6L**
> > > > > >
> > > > > > Dear Reviewer Si6L,
> > > > > >
> > > > > > We are deeply grateful for both your appreciation of our work and the constructive feedback you provided. We will revise our paper in response to your insightful suggestions and strive to ensure an even greater impact in our subsequent work. Thank you once again for your precious time and thoughtful review.
> > > > > >
> > > > > > Best regards,
> > > > > >
> > > > > > Paper 13149 Authors

---

### Official Review · Reviewer_BGbw · 2025-07-02

**Clarity:** 3
**Significance:** 3
**Originality:** 3
**Rating:** 4
**Confidence:** 3

**Summary:**

The paper introduces RG-VLM, an MLLM model that relies on reinforcement learning in visual reasoning generation process. The authors propose Reasoning Generation Policy Optimization (RGPO) algorithm to effectively guide multimodal policy update. RG-VLM is compared with several counterpart approaches, showing its clear reasoning advantage.

**Questions:**

1. Are there any other limitations of RG-VLM besides the one mentioned in Appendix A3?
2. It will be interesting to see how effective would RG-VLM be in solving „classical” abstract reasoning tasks, such as Raven’s Progressive Matrices or Bongard Problems.
3. How the RG-VLM Chain-of-Thought reasoning conceptually differs from that of DeepSeek?

**Ethical Concerns:**

["NO or VERY MINOR ethics concerns only"]

**Final Justification:**

My concerns are mostly addressed, however, I still believe RG-VLM requires more extensive experimental evaluation on other types of reasoning tasks to confirm its universality and overall strength.

**Limitations:**

Discussion of limitations is extremely brief (just one sentence).

**Paper Formatting Concerns:**

The paper begins with the figure (Figure 1) placed above the abstract, which is highly unusual.

**Quality:**

3

**Strengths And Weaknesses:**

Strengths:
1. Introduction of RG-VLM, and MLLM dedicated to solving visual reasoning tasks and showing strong performance across diverse benchmark datasets.
2. Proposal of RGPO algorithm that has proven very effective in the Chain-of-Thought reasoning.
3. The research is very well motivated and the main ideas are clearly presented.

Weaknesses:
1. The experimental evaluation completely omits abstract visual reasoning benchmarks, i.e. Raven’s Progressive Matrices, Bongard Problems, Visal Analogy Problems, and other.
2. A discussion on RG-VLM limitations is extremely brief.

---

> ### Author Rebuttal · Authors · 2025-07-31
>
> We sincerely appreciate the reviewer for the constructive comments and will address the concerns raised.
>
> **W1&Q2: Abstract visual reasoning benchmark.**
>
> **Ans:**
> We appreciate the reviewer’s valuable suggestion. While our work mainly centers on reinforcement learning strategies for image reasoning generation, our dual KL approach ensures that model performance does not deviate significantly from the original policy. Accordingly, as shown in Table 14, our RG-VLM performs similarly to Qwen2.5-VL on the visual reasoning benchmark mentioned by the reviewer.
>
> |      **Model**    | **Raven** | **Bongard Problems** | **Visal Analogy Problems** |
> |-------------------|:---------:|:---------:|:--------------:|
> | Qwen2.5-VL        |    0.10   |   0.16    |       0.193    |
> | RG-VLM (ours)     |    0.10   |   0.15    |       0.184    |
>
> **Table 14: Training GPU Hours comparison for unified models.**
>
> **W2&Q1: Discussion on RG-VLM limitations is extremely brief.**
>
> **Ans:**
> An additional limitation at present lies in the reward function. The CLIP-score-based consistency reward does not adequately capture low-level image details or numerical information. We plan to further explore this in future work.
>
>
> **Q3: Conceptual differences between RG-VLM's and DeepSeek's Chain-of-Thought reasoning.**
>
> **Ans:**
> RG-VLM is capable of both multimodal input and output, whereas DeepSeek is limited to textual processing. Moreover, the Chain-of-Thought (CoT) in RG-VLM serves as a reasoning mechanism to enhance visual generation, while DeepSeek’s CoT is designed solely for textual reasoning tasks.

---

> > ### Comment · Reviewer_BGbw · 2025-08-04
> >
> > Thank you for the rebuttal. My concerns are mostly addressed, however, my comment about the efficacy of RG-VLM in solving „classical” abstract reasoning tasks, such as Raven’s Progressive Matrices or Bongard Problems is answered in the number of training hours, which does not say much about the model’s performance on these tasks.

---

> ### Author Response · Authors · 2025-08-04
> **Response to the official comment by reviewer BGbw**
>
> Dear Reviewer BGbw,
>
> Thanks for your response. We are glad that our rebuttal is able to address your concerns. We sincerely appreciate your time and effort in reviewing our paper. Your valuable comments are crucial in improving the quality of our work.
>
> Additionally, we apologize for the typo in the caption of Table 14. This table simply presents the performance results for your listed tasks. The revised version is "Table 14: Performance of visual reasoning tasks" accordingly. We followed the methodologies outlined in [1][2][3] to evaluate these three benchmarks, and our results indicate that our model performs comparably to the Qwen2.5-VL (the expert model for visual understanding), benefiting from our dual-KL strategy in the RL stage.
>
> Best regards,
>
> Paper 13149 Authors
>
> [1] https://github.com/apple/ml-rpm-bench
>
> [2] Wüst, Antonia, et al. "Bongard in Wonderland: Visual Puzzles that Still Make AI Go Mad?." arXiv preprint arXiv:2410.19546.
>
> [3] Bi, Jing, et al. "Verify: A benchmark of visual explanation and reasoning for investigating multimodal reasoning fidelity." arXiv preprint arXiv:2503.11557.

---

> > ### Comment · Reviewer_BGbw · 2025-08-04
> >
> > Thank you for clarifying the content of Table 14. Apparently, in RPM, VAP and BP the proposed approach does not offer any performance advantage compared to Qwen2.5-VL. I believe RG-VLM requires more extensive exprimental evaluation on other problems to confirm its universality and overall strength. Therefore, I am lowering my score to 4: Borderline accept.

---

> ### Author Response · Authors · 2025-08-05
> **Further response to reviewer BGbw**
>
> Dear Reviewer BGbw,
>
> Thanks for your response. It appears there may have been some misunderstanding. The core contribution of our work is the proposed reinforcement learning strategy, which is specifically designed for reasoning generation tasks, rather than visual understanding. As demonstrated in the ablation study in Table 5 of the manuscript, this RL strategy improves the generation performance of our unified VLM by 6% points.
>
> Moreover, Table 14 presents zero-shot test results on the visual reasoning tasks of our model trained with only image generation data. To further clarify our proposed RL strategy, we included only 1k relevant visual understanding samples during the RL stage, enabling us to further assess the model's performance on these tasks as shown in Table 16. Our initial goal is to use RL to transfer the reasoning capabilities of VLM to generation. However, this training method can also be fed back to VLM to improve understanding ability.
>
> If you still have any concerns, we would be pleased to discuss them further with you!
>
>
> |    |   **Model**    | **Raven** | **Bongard Problems** | **Visal Analogy Problems** | **Overall** |
> |----|:---------------|:---------:|:---------:|:--------------:|:----:|
> | Only Und. (Expert Model)| InternVL        |    0.10   |   0.15    |       0.183    | 0.144 |
> | | LLaVA-1.5        |    0.09   |   0.14    |       0.176    | 0.135|
> | | Qwen2.5-VL        |    0.10   |   0.16    |       0.193    | 0.151 |
> | Und. & Gen. (Unified Model)| Janus-Pro     |    0.07   |   0.12    |       0.154    | 0.115 |
> | | Bagel     |    0.09   |   0.13    |       0.182    | 0.134|
> | | RG-VLM (ours)     |    0.10   |   0.15    |       0.184    | 0.145 |
> | | RG-VLM* (ours)    |    **0.11**   |   0.15    |       **0.210**    | **0.157** |
>
>
> **Table 16: Detailed performance comparison. * indicates the model with more 1k visual understanding training data in the RL stage. Und. and Gen. denote understanding and generation, respectively.**
>
> Best regards,
>
> Paper 13149 Authors

---

> > ### Comment · Reviewer_BGbw · 2025-08-05
> >
> > Thank you for your clarification

---

### Official Review · Reviewer_Joq5 · 2025-07-03

**Clarity:** 3
**Significance:** 3
**Originality:** 2
**Rating:** 4
**Confidence:** 4

**Summary:**

The paper proposes RG-VLM, a unified VLM.  RG-VLM attaches an OmniGen diffusion decoder to a Qwen-2.5 backbone via lightweight connectors and fine-tunes it with Reasoning-Generation Policy Optimisation (RGPO): an RL scheme that extends GRPO with a CLIP-based “consistency” reward and dual KL regularisers. RG-VLM delivers SoTA or near-SoTA scores on six benchmarks spanning multimodal understanding (MMMU, MMBench, RealWorldQA) and reasoning-centric generation (GenEval, DPG-Bench, WISE), while preserving baseline comprehension performance.

**Questions:**

The output images from RG-VLM appear to have simpler backgrounds, saturated colors, and reduced aesthetic appeal. Could this be due to the reward design? What might improve this?

**Ethical Concerns:**

["NO or VERY MINOR ethics concerns only"]

**Final Justification:**

I appreciate the added experiments, but concerns about color saturation and aesthetics remain. No concrete evidence shows improved color fidelity or appeal. I keep my rating.

**Limitations:**

The limitations described in the paper seem more like future work. Insufficient attention is paid to the real limitations.

**Quality:**

3

**Strengths And Weaknesses:**

Strengths:
- Brings chain-of-thought (CoT) reasoning into image generation/editing, advancing multimodal agent capabilities.
- RGPO’s extension over GRPO is well documented; ablations (Tables 6-9) isolate each reward/regulariser’s effect.
- Six quantitative benchmarks plus qualitative comparisons with GPT-4o and Gemini 2.5 demonstrate versatility.
- Appendices list datasets, hyper-parameters, and training stages, making replication easier.

Weaknesses:
- RGPO adds a CLIP reward and dual KL loss to GRPO; similar CLIP-guided RL appeared in SimpleAR (2025), not much has changed conceptually.
- All alignment metrics are CLIP-based; no user study confirms that generated images faithfully reflect the model’s textual reasoning.
- Consistency reward is computed on low-resolution latents, potentially encouraging adversarial CLIP features rather than genuine semantics.
- The number of available examples is limited. Including more would help reinforce the paper's claims.

---

> ### Author Rebuttal · Authors · 2025-07-31
>
> We sincerely appreciate the reviewer for the precious time and valuable comments and will address the concerns raised.
>
> **W1: The difference from SimplerAR​.**
>
> **Ans:**
> Our contributions lie in two key areas: (1) constructing a unified VLM capable of both understanding and generation, and (2) leveraging reinforcement learning to unleash reasoning generation. To the best of our knowledge, SimpleAR and our work are conducted concurrently. But, unlike SimpleAR, which is an image generation expert method and employs a single KL constraint, our method supports both understanding and generation, as well as integrates a dual-KL formulation. We have included comprehensive ablation studies as shown in Tables 7-9 of the manuscript to validate its effectiveness within a unified architecture.
>
> **W2: Lack of user studies.**
>
> **Ans:**
> We conduct a user study as shown in Table 12 below, with 50 participants following the previous method. Participants are required to select the most preferred results from a set of methods, including only-support generation, FLUX.dev, and the unified understanding and generation counterparts, Janus-Pro and Bagel.
>
> | **Method** | **User Study** |
> |------------|:------------:|
> | FLUX.dev  | 0.32 |
> | Janus-Pro  | 0.10 |
> | Bagel | 0.28 |
> | RG-VLM (ours) | 0.30 |
>
> **Table 12: User Study.**
>
> **W3: Questions about consistency reward.**
>
> **Ans:**
> Potential misunderstandings may arise here. In the third stage, our consistency reward is calculated on the generated RGB image rather than the low-level latent representation, as detailed in Section 3.4 of the manuscript. Additionally, we conduct ablation studies to evaluate the effects of different reward models under consistent settings as shown in Tables 7-9 of the manuscript. Switching to a VLM-based reward model directly offers no substantial advantage compared to the CLIP model. We will further investigate the role of the reward model in the reinforcement learning process in our future work.
>
> | **Reward Model** | **WISE Overall Score** |
> |------------|:------------:|
> | CLIP-ViT  | 0.598 |
> | VisualPRM  | 0.606 |
>
> **Table 13: Ablation of different reward models.**
>
> **W4: The Number of available examples is limited.**
>
> **Ans:**
> We have presented some text-controlled image generation results in Figure 7 in the appendix part of the manuscript, but due to rebuttal policy limitations, we cannot provide more at this rebuttal period. We will release our model publicly and update the revised version with more diversity visualizations.
>
> **Q1: The output images have simpler backgrounds.**
>
> **Ans:**
> If the reviewer is referring to the example in Figure 1, the observed content stems from the model-generated CoT, which includes certain stylistic descriptions, which are illustrated in Figures 8–10 in the appendix. Our model exhibits strong prompt-following capabilities, which likely accounts for this phenomenon. By tuning the temperature and sampling configurations of the VLM, one can control the variability of the generated outputs. What's more, additional results of text-controlled generation are presented in Figure 7.
>
> **Q2: Insufficient attention is paid to the real limitations.**
>
> **Ans:**
> The primary limitation at present lies in the reward function. The CLIP-score-based consistency reward does not adequately capture low-level image details or numerical information. We plan to further explore this direction in future work.

---

### Official Review · Reviewer_s7fx · 2025-07-03

**Clarity:** 3
**Significance:** 3
**Originality:** 3
**Rating:** 5
**Confidence:** 3

**Summary:**

The paper introduces RG-VLM, a unified multimodal large language model that advances reasoning generation in vision-language tasks by integrating reinforcement learning via the Reasoning Generation Policy Optimization (RGPO) algorithm. The approach involves a three-phase training strategy: the creation of a decoder-only diffusion-based unified architecture, supervised fine-tuning using Chain-of-Thought (CoT) instruction data, and further training via RL with multimodal feedback rewards. RG-VLM demonstrates strong performance in both image understanding and generation tasks, outperforming recent unified models on established benchmarks, and excels on complex reasoning and editing scenarios. The paper includes empirical evidence from several public benchmarks and qualitative results across a spectrum of vision-language scenarios.

**Questions:**

How does RG-VLM compare (quantitatively) with the very latest closed and proprietary models (e.g., Gemini-2.5, GPT-4o) on publicly available benchmarks, given that only qualitative results (Figure 6) are provided? Are there any plans or ongoing work to provide more direct numerical comparisons with these models?

**Ethical Concerns:**

["NO or VERY MINOR ethics concerns only"]

**Final Justification:**

I appreciate the authors providing a thorough response to my concerns. Most of my concerns have been addressed.

**Limitations:**

yes

**Paper Formatting Concerns:**

no formatting issue

**Quality:**

3

**Strengths And Weaknesses:**

Strengths:
1. Reinforcement Learning for Reasoning Generation: The integration of RGPO, a reinforcement learning algorithm with multimodal feedback and explicit Chain-of-Thought supervision, is well-motivated and distinguishes RG-VLM from models relying solely on supervised or prompt-based approaches. The reward shaping strategy is explained in detail, offering insights into how multimodal consistency and CoT structure are operationalized during policy optimization.
2. Image Editing/Generation with Reasoning: Figure 1 and Figure 6 visually demonstrate RG-VLM’s strengths in both plain text reasoning (with CoT) and complex image editing/generation. For instance, Figure 1’s concrete reasoning scenarios and vision editing examples illustrate capabilities not present in competitor models.
Weaknesses:
1. Limited Multimodal CoT: The current CoT process is stated (Appendix A.3, but addressed in main text as well) to operate only in plain text; true multimodal reasoning chains (i.e., with explicit image intermediate steps or edits) are left for future work. This constrains the novelty and applicability of the reasoning generation claim, as current generation is essentially textual CoT guiding generation, rather than fully realized multimodal reasoning sequences.

---

> ### Author Rebuttal · Authors · 2025-07-31
>
> We sincerely appreciate the reviewer for the time and comments and will address the concerns raised.
>
> **W1: Limited multimodal CoT.**
>
> **Ans:**
> Our work aims to move beyond the limitations of conventional fine-tuning and prompt rewriting strategies, which often fail to fully harness the reasoning potential of large models due to incorrect text guidance or inefficient prompt following execution. We introduce a reinforcement learning-based approach that unleashes the reasoning generation capabilities under our developed unified VLM. While current CoT reasoning remains text-based, we have shown it can substantially improve performance, laying a foundation for future multimodal CoT extensions research.
>
> | **Method** | **WISE Overall Score** |
> |------------|:------------:|
> | Baseline  | 0.44 |
> | Prompt-Rewrite by Qwen2.5  | 0.49 |
> | RG-VLM (w/ CoT) | 0.61 |
>
> **Table 10: Effectiveness of the CoT of our model.**
>
> **Q1: Closed models' performance.**
>
> **Ans:**
> We perform iterative training using open-source data and evaluate GPT-4o on the WISE benchmark. As shown in Table 11, our model significantly narrows the performance gap with closed-source counterparts compared to open models.
>
> | **Method** | **WISE Overall Score** |
> |------------|:------------:|
> | Janus-Pro  | 0.35 |
> | MetaQuery-XL  | 0.55 |
> | RG-VLM (w/ CoT) | 0.71 |
> | GPT-4o | 0.80 |
>
> **Table 11: Performance Comparison.**

---

> > ### Comment · Reviewer_s7fx · 2025-08-03
> > **The response solve most of my concerns**
> >
> > I appreciate the authors providing a thorough response to my concerns. Most of my concerns have been addressed.

---

> ### Author Response · Authors · 2025-08-04
>
> Dear Reviewer s7fx,
>
> Thanks for your response. We are glad that our rebuttal is able to address your concerns. We sincerely appreciate your review time and the valuable comments. We will attach great importance to revising the paper according to your suggestions.
>
> Best regards,
>
> Paper 13149 Authors

---

### Note · Authors · 2025-08-12

Dear ACs,

We sincerely thank all reviewers and ACs for their effort and precious time. We are writing this letter to provide a comprehensive summary of this rebuttal.

We are glad that the proposed concerns of reviewers have been addressed and are deeply honored that all reviewers have expressed their willingness to accept our RG-VLM. Our RG-VLM is recognized as: the paper is “well-written and represents a very interesting exploration of unified MLLMs in the domain of reasoning.” (Reviewer Si6L), the method is “very well-motivated and offers insights”(Reviewer s7fx, Reviewer BGbw), the experiments “sufficiently validate the effectiveness of the proposed method” (Reviewer Joq5), and the performance “is promising”. We will attach great importance to incorporating all the constructive suggestions from all reviewers in the revised version, and sincerely thank the reviewers for their valuable comments once again.

At last, we thank again for ACs' effort in processing this paper.

Best regards,

Paper 13149 Authors

---

### Decision · Program_Chairs · 2025-09-17

**Decision:**

Accept (poster)

**Comment:**

This work proposes a unified visual understanding and generation framework and enhances the reasoning capacity of the model by reasoning generation policy optimization (RGPO) algorithm. The study is well motivated, and the empirical result is impressive. All reviewers have positive scores and concerns, such as the acquisition of Chain-of-Thought abilities, and the details of the RLVR reward, were addressed after rebuttal. Overall, the work demonstrates a promising direction. Please incorporate the comments from all discussions and improve the revision with more extensive experimental evaluation as suggested, e.g., different reasoning tasks, complex background, etc.